# Changing Antibiotic Prescribing Cultures: A Comprehensive Review of Social Factors in Outpatient Antimicrobial Stewardship and Lessons Learned from the Local Initiative AnTiB

**DOI:** 10.3390/antibiotics14111068

**Published:** 2025-10-24

**Authors:** Janina Soler Wenglein, Reinhard Bornemann, Johannes Hartmann, Markus Hufnagel, Roland Tillmann

**Affiliations:** 1Department of Pediatrics, Protestant Hospital of the Bethel Foundation, Medical School and University Medical Center OWL, Bielefeld University, 33617 Bielefeld, Germany; 2Department of Population Medicine and Health Services Research (AG2), School of Public Health, Bielefeld University, 33615 Bielefeld, Germany; bornemann@uni-bielefeld.de; 3Gemeinschaftspraxis Hausärztliche Internisten, Ärztenetz Bielefeld, 33602 Bielefeld, Germany; kontakt@praxis-eckardtsheim.de; 4Department of General Pediatrics and Adolescent Medicine, Center for Pediatrics, Division of Pediatric Rheumatology and Clinical Infectious Diseases, Medical Center and Medical Faculty, University of Freiburg, 79106 Freiburg, Germany; markus.hufnagel@uniklinik-freiburg.de; 5Praxis für Kinder- und Jugendmedizin, Ärztenetz Bielefeld, 33602 Bielefeld, Germany

**Keywords:** antimicrobial resistance, antibiotic stewardship, antimicrobial stewardship, antibiotic prescribing, best-practice, outpatient

## Abstract

Antimicrobial resistance (AMR) constitutes a major global health challenge, driven significantly by inappropriate antibiotic use in human medicine. Despite the existence of evidence-based guidelines, variability in antibiotic prescribing persists, influenced by psychosocial factors, diagnostic uncertainty, patient expectations, and local prescribing cultures. Outpatient care, the setting in which most antibiotics are prescribed, is particularly affected by such challenges. Traditional top-down interventions, such as national guidelines, often fail to achieve sustained behavioral change among prescribers. In this comprehensive review, we provide an overview of the psychological and behavioral factors influencing antimicrobial stewardship (AMS) implementation, as well as describe a bottom-up project working to meet these challenges: the “Antibiotic Therapy in Bielefeld” (AnTiB) initiative. AnTiB employs a cross-sectoral strategy aimed at developing rational prescribing culture by means of locally developed consensus guidelines, interdisciplinary collaboration, and regularly held trainings. By addressing both the organizational and psychological aspects of prescribing practices, AnTiB has facilitated a harmonization of antibiotic use across specialties and care interfaces at the local level. The initiative’s success has led to its expansion within Germany, including through the creation of the AMS-Network Westphalia Lippe and the development of AnTiB-based national pediatric recommendations. These projects are all grounded in social structures designed to strengthen the long-term establishment of AMS measures. Our efforts underscore the importance of considering local social norms, professional network, and real-world practice conditions in AMS interventions. Integrating behavioral and social science approaches into outpatient antimicrobial stewardship—exemplified by the practitioner-led AnTiB model—improves acceptability and alignment with stewardship principles; wider adoption will require local adaptation, routine outpatient resistance surveillance, structured evaluation, and sustainable support.

## 1. Introduction

Antimicrobial resistance (AMR) is a global health threat driven by diverse factors, critical among them are the use of antibiotics (AB) in human and veterinary medicine [1,2]. These circumstances are particularly fueled by the overprescription and inappropriate use of AB at large and of specific antibiotic substances in detail. Therefore, optimizing antibiotic use, alongside other interventions such as AMR screening and hygiene measures, is crucial for controlling AMR.

In this respect, broad variations in global and local antibiotic prescribing practices have been observed for a long time, both in connection with the number of prescriptions given (a quantitative measurement) and in relation to AB misuse, especially of antibiotics with a broader spectrum (a qualitative measurement). These variations have been documented in multiple dimensions: by geographic regions, according to inter-sectoral differences (inpatient vs. outpatient care), and among medical specialties. For example, significant disparities exist in antibiotic consumption between southern and northern European countries [3], as well as within specific countries [4]. Within Germany, AB prescribing rates in the eastern and southern federal states are lower, while higher rates persist in the western areas of the country, a phenomenon that remains insufficiently understood [5,6]. Interestingly, these rates are similar for both children and adults, i.e., variations cannot be fully explained by socioeconomic factors or healthcare density [5,7]. However, qualitative research data has suggested that low-prescribing areas benefit from strong collegial networks, effective laboratory collaborations, and low numbers of patient requests for antibiotic prescriptions. By contrast, physicians in high-prescribing regions are more likely to report poor coordination between healthcare services, limited access to information on antimicrobial resistance, and fewer professional development opportunities [7]. These differences also may significantly depend on standards of available medical care in a given region.

Given increasing rates of AMR, efforts have intensified in attempts to improve prescribing quality. After the initial targeting of this goal approximately 20 years ago, a variety of intervention measures were designed and implemented. One example is the “Choosing Wisely” campaign [8], which helped define quality indicators and training targets for healthcare professionals, as well as establish teams to supervise in-hospital antibiotic prescriptions and provide infectious diseases consultations [9,10]. In recent years, these efforts have been described under the umbrella term of “antimicrobial stewardship” (AMS) or antibiotic stewardship (ABS).

To date, AMS activities have predominantly included “classical” approaches such as a multi-level medical education, the professional training of practicing physicians, and the publication of journal articles and recommendations/guidelines. However, merely publishing such guidelines does little to ensure that prescribing practices among healthcare professionals will be affected; in fact, the publication of guidelines frequently fails to significantly influence behavior [11]. Guidelines may lead to short-term benefits, but only rarely have been shown to have long-lasting effects on actual AB prescribing [12]. These observations, which were first discussed over a decade ago, led to the insight that guidelines and other “technical” strategies missed the opportunity to examine the role that other factors might play in shaping AB prescribing practices—namely “local prescribing cultures” or “prescribing etiquette” [13,14,15].

Innovatively, this “socio-cultural” concept focused not just on the distribution of scientific information (guidelines, etc.) but also on factors immediately relevant to the delivery of medical care, including, for example, an examination of personal beliefs about antibiotic efficacy, clinical experience at large, case studies with negative impacts, adherence to guidelines, and more. As one example, in various medical specialties, (national) guidelines aim to propagate the best available scientific evidence; however, practitioners on the forefront of delivering care often perceive these guidelines as disconnected from the realities of their daily clinical routines and the implementation challenges that these can pose. Diagnostic uncertainty can further complicate the picture, and patient expectations regarding which prescriptions they should be receiving may contribute to overprescribing as well.

To bridge these gaps, the “socio-cultural” approach to combatting AMR has been identified as crucial to the development of strategies for improving the integration of scientific evidence into everyday medical practice, while also taking pragmatic matters into consideration, including, for example, organizational circumstances. This approach further suggests that the strategies pursued, or “bridges,” need to operate in a bidirectional manner, ensuring not only the “top-down” transmission of scientific information but also the incorporation of “bottom-up” feedback from everyday medical care. Rather than a “top-down” dissemination of guidelines, the “bottom-up” process used in “Antibiotic Therapy in Bielefeld” (AnTiB) involves the following: (1) convening local practitioner groups; (2) developing consensus recommendations based on clinical experience, literature review, and expert input; (3) piloting and adapting recommendations to local workflows; and (4) maintaining the recommendations through regular practitioners’ feedback and updates.

In the comprehensive narrative review presented here, we offer an overview of the literature related to “contextual” factors influencing antibiotic prescribing, as well as an in-depth profile of the local AMS project AnTiB, which was founded with the idea of focusing on AMS at a practical, local level. Our literature search targeted AMS interventions employing contextual/socio-cultural strategies that had as their goal the improvement of antibiotic prescribing behaviors.

## 2. Contextual Factors Related to Antibiotic Prescribing

Social norms—the mostly unspoken and unconscious “rules” for what may be considered appropriate or inappropriate within a society or a group—significantly influence personal and group behavior. In setting expectations for behavior, they help maintain social order and cohesion and, as such, may be a key factor influencing antibiotic prescribing [16,17]. In addition to the organizational and individual [18] strategies already described, numerous effective AMS interventions have leveraged behavioral strategies by capturing social references for prescribers, informing them, for example, about their either high [19] or low prescription rates [20]. Additional factors influencing prescribing behavior, like organizational factors and individual factors [18], have been outlined above.

Time pressure, or lack of time, has frequently been identified as a factor affecting or directing human decision-making. This has been observed in multiple contexts, not just in areas related to AMS: time pressure can (1) make people sacrifice accuracy or thoroughness when completing tasks [21], (2) make people more risk-averse (by encouraging a focus on minimizing negative outcomes) [22], (3) promote the use of simpler decision-making strategies (i.e., heuristic [23] or intuitive, rather than analytical), (4) elevate negative information [24], and (5) support the repetition of choices previously made [21].

It is unsurprising then that time pressure would have an impact on the delivery of outpatient care where, importantly, a high throughput of patients in short time periods is common. In addition, different prescribing styles, social norms, and individual behaviors can lead to divergent prescribing patterns. When different types of care interface, such as during emergency services or at the transition between outpatient and inpatient care settings, the variability in prescribing practices that follows can become particularly problematic, which not unfrequently results in conflict, uncertainty, and loss of trust between patients and healthcare providers.

Addressing such challenges requires a coordinated approach, one that extends beyond traditional approaches that singularly emphasize the dissemination of medical information as the path to success. Instead, recent approaches emphasize the importance of harmonizing communication and developing more pragmatic strategies to support improved antibiotic prescribing practices. This is particularly important at the local level, where most care interfaces occur. The literature suggests that AMS interventions incorporate strategies that emphasize the psychological impact of learning from role models and of adhering to social norms [13,14]. In doing so, greater certainty in prescribing practices and a reduction in inconsistencies can be encouraged, thereby also enhancing trust and supporting positive care outcomes. To achieve this, multiple reports say AMS efforts need to be implemented both within specific medical specialties and in an interdisciplinary manner, crossing both sectors and institutional boundaries. The participation of prescribers has been shown to be especially important [25].

Challenges and barriers in implementing AMS programs are multifactorial; they include patient communications, practitioner expectations, practitioner resistance, organizational limitations, and resource constraints. In regions where antibiotics are prescribed at low rates, supportive contextual factors such as strong collegial networks, good collaborations with hospitals, and responsive microbiological laboratories have all been identified as important resources [7]. Therefore, interventions targeting professional development, improved collaboration structures, best-practice guidelines, and better networking opportunities for physicians may potentially support more rational antibiotic prescribing behavior [7]. As described by the social contagion theory, social environment significantly influences behavior. This suggests that behaviors, emotions, and attitudes can be transmitted within a group and can become reinforced through interactions among individuals [26]. Hence, it is not just infections and antibiotic resistance that may be contagious but also attitudes toward medical practice, including, in this case, antibiotic prescribing practices [25]. But this can be taken as a positive as well, because ABS may also be considered to have “infectious” potential—in a good sense. It is simply a matter of finding the right vector for this “infection”: implicit social rules transformed into explicit guidelines that gain acceptance by means of social factors, in parallel with the integration of up-to-date scientific standards.

## 3. Outpatient AMS Initiatives—The AnTiB Example

In Germany, approximately 85% of antibiotics in humans are prescribed in outpatient settings. This suggests there is a significant need for effective AMS programs targeted at improving antibiotic prescribing in this area. Although regional outpatient AMS initiatives and programs in Germany actively operate in diverse healthcare settings [27], they often focus on specific measures targeting individuals (e.g., the education of medical professionals), rather than attempting to address local “prescribing cultures” at large. Monitoring systems such as audits or feedback reports, along with compliance tracking and feedback systems [28,29,30,31], are often reported as outpatient AMS strategies [32,33,34]. Education programs for clinicians, patients [29], staff, and providers [28] are also frequently mentioned. Described in the context of their direct educational function are handbooks, national and international guidelines, and infectious disease specialist guidance, as well as communications [31,32,35].

In response to the problems resulting from the high variance in antibiotic prescribing at the care interface in outpatient settings in Germany in 2016, the AnTiB initiative was established by the Ärztenetz Bielefeld. The Ärztenetz Bielefeld is a local network of specialist physicians providing outpatient care, along with general practitioners (GPs) and pediatric GPs. Bielefeld, a city with approximately 330,000 inhabitants, is located in the federal state of North Rhine-Westphalia (NRW) in Germany. AnTiB is a practitioner-led, bottom-up outpatient antimicrobial stewardship (AMS) initiative. Its primary aims are (1) to harmonize outpatient antibiotic prescribing through locally consented, pragmatic recommendations for common infections; (2) to improve communication across outpatient–inpatient interfaces; (3) to build interdisciplinary networks (general practitioners, pediatricians, hospital clinicians, microbiologists, pharmacists); and (4) to implement regular training, audit and feedback, and easily accessible decision aids (e.g., pocket cards). The development process combined open invitations with relevant clinicians, structured consensus meetings, external expert review, pilot testing in routine care, and iterative updates based on user feedback and updated evidence. The AnTiB initiative was launched at a meeting of local outpatient pediatricians interested in organizing a symposium on the subject of AMS, as well as in AMS more generally. Through their discussions, shortcomings in local outpatient AMS became apparent, but significant untapped opportunities for improved cooperation also seemed clear. An early goal was simply to establish communication about AMS issues among local practitioners at the frontline of outpatient care, both in pediatric GP offices and at the outpatient clinic of the local children’s hospital. Then, as a first concrete step, these doctors established the “AnTiB Paed 2017” project. The project’s innovative aim was to compile recommendations for pediatric antibiotic prescribing by taking a “bottom-up” rather than (conventional) “top-down” approach. This has since become considered an exemplary model.

The recommendations thus generated grew out of a structured process [36] that involved approximately 25 outpatient pediatric GPs plus physicians from the local pediatric hospital. This collaboration between medical providers working in differing settings (outpatient and inpatient) later proved key to having practitioners accept the guidelines created: equal participation and input had been built into the guideline development process from the beginning. The process was initiated through a formalized invitation welcoming all relevant outpatient physicians and emergency service providers to participate. In addition, specific interfaces for managing communication between the hospital and outpatient settings were identified, and both structured exchange processes and contact networks were established. For outpatient pediatrics, a consensus on standard treatments for the most common infectious diseases was developed, grounded in evidence-based scientific guidelines that had already been peer-reviewed by external experts. By reducing complexity and eliminating contradictions, as well as by offering accessible guidance, the approach facilitated a fast and relatively uncomplicated process that produced results. The final consensus was later formalized into a user-friendly publication to assist medical decision-making.

The consensus recommendations produced by the project aim to reduce antibiotic prescriptions and are based on simple AMS principles, exemplified by the following pediatric antibiotic prescribing recommendations: (1) avoid unnecessary antibiotic therapy (or stop it immediately); (2) keep necessary antibiotic therapy as short and narrow-spectrum as possible; (3) avoid antibiotic therapy for mild, self-limiting bacterial diseases in immunocompetent patients; (4) in unclear situations without a risk constellation, wait (known as “watchful waiting”); (5) reduce topical antibiotic therapy, e.g., for skin and eye infections; (6) improve the quality of AB prescriptions by specifying dose, duration of therapy and conditions of use on the prescription itself; (7) reduce critical antibiotics and only use them selectively (e.g., oral cefuroxime due to poor oral bioavailability and potential development of multi-drug-resistant Gram-negative bacteria (MRGN)).

The AnTiB initiative quickly spread and became adopted by other medical specialties in Bielefeld, including gynecology, general GP (adult) medicine, urology, and otorhinolaryngology (ENT), all of which similarly initiated cooperations within their own professional groups. Soon afterwards, each also issued its own antibiotic recommendations following the process described above. Participants were asked to regularly take part in surveys to evaluate the implementation of the recommendations and their own satisfaction with them. Survey feedback showed participants to have been highly satisfied with the applicability and practicability of the locally developed recommendations (unpublished data).

At the same time, these locally developed recommendations continued undergoing revisions in order to keep up with the latest scientific developments [36]. Following this “core activity” of the AnTiB project, a variety of other local AMS activities began to emerge [37]. For a timeline of the activities developed, see Figure 1.

Regarding the psychosocial aspects of implementing AMS interventions, AnTiB’s first aim was to achieve this by focusing on establishing a local “prescribing culture” in outpatient settings, as well as within the different medical disciplines involved. As a second goal, it sought to harmonize the intersection of outpatient–inpatient care by involving the various hospital departments and medical specialty groups. Third, AnTiB targeted other professional groups relevant to the AMS field, mainly microbiologists, laboratories, and pharmacists.

## 4. From the Local AnTiB Project to the Regional “AMS Network Westphalia-Lippe” and Beyond

By 2018, a broader AMS network in the Bielefeld area, which had expanded to the surrounding district, had begun to form. In 2022, several regional AMS stakeholders formally established the “ABS-Network Westphalia-Lippe”, which now covered nearly half of the residents (8.3 of 17.9 million) of the federal state of NRW. This network comprises and/or is supported by GP networks, laboratory physicians (especially microbiologists), pharmacists, hospitals, regional hospital physicians, and physicians working in outpatient care, as well as by a number of health authorities [38]. The network cooperates with other networks of healthcare professionals, provides a communication platform, offers free online and face-to-face training on AMS topics, integrates AMS principles into university teaching activities, and conducts public relations work around AMS—all significant accomplishments achieved in a short time frame.

Both the AnTiB project and the ABS-Network Westphalia-Lippe primarily aim to change local prescribing habits and reduce diagnostic and therapeutic uncertainty by aligning routine practice with antibiotic stewardship principles. To achieve this, the project promotes locally adapted consensus guidance, easy-to-use decision support, and accessible best-practice information, together with education and quality improvement measures. They do so by embracing the power of social learning and imitation and cognitive biases, as well as by taking “real-life” circumstances of care seriously. Existing social groups, including doctors’ networks, are used to disseminate information and encourage participation.

But the efforts have not stopped here. The initial AnTiB pediatric activities were practitioner-led at a local level. Subsequently, these activities contributed to the formation of the Working Group Antibiotic Stewardship in Pediatrics (ABSaP) under the auspice of the German Society for Pediatric Infectious Diseases (DGPI). The working group comprises the balanced number of scientists and pediatric practitioners, focuses on topics such as diagnostic stewardship and the de-labeling of antibiotic allergies, and works to ensure that evidence-based practices are both feasible and widely accepted in pediatric care [39]. Through ABSaP/DGPI, the locally developed recommendations were refined, peer-reviewed, and disseminated at national and international levels [39]. AnTiB aligns the pediatric AnTiB recommendations with the DGPI recommendations and reviews local recommendations during each revision of national recommendations for possible locally necessary deviations and consistency. Meanwhile, AnTiB has also become affiliated with the Professional Association of Pediatricians (BVKJ) in Germany and other infectious disease specialists nationwide.

The BVKJ’s commitment to pediatric AMS highlights the impact of the original AnTiB initiative, especially as the BVKJ’s members are primarily practitioners working in non-hospital outpatient settings. In fact, the BVKJ has designated AMS as a central theme for 2025. Further, the BVKJ recently launched a comprehensive, continuously running, nationwide training program tailored to meet the needs of pediatric healthcare providers and designed to embed AMS principles into daily practice. The AnTiB concept, initially a local project, has now been rolled out to pediatric departments across Germany—a successful implementation of its own recommendations. The working group’s guidelines are consistently disseminated through diverse media platforms. Their relevance and applicability to pediatric healthcare practice is always emphasized. This integrated approach not only elevates the overall standard of pediatric care but also ensures a sustainable positive impact on public health outcomes by fostering a culture of responsible antibiotic use within the pediatric care community.

Together with the Association of Statutory Health Insurance Physicians Westphalia-Lippe (Kassenärztliche Vereinigung Westfalen-Lippe, KVWL), the AnTiB project established a prescription feedback system for outpatient practices. Evaluations are regularly performed by means of cooperation with the School of Public Health at Bielefeld University. Press releases are put out on a regular basis, e.g., for the annual World AMR Awareness Week celebrated by the WHO. The original local AnTiB approaches are all integrated and further developed within the regional ABS-Network Westphalia-Lippe [40]. Additionally, continuous research plays a key role in local and regional AMS activities, with numerous publications contributing to the initiative’s visibility across Germany and perhaps beyond. To make the project aims explicit, AnTiB adopted core stewardship elements aligned with international frameworks: (a) accountability; (b) locally adapted, pragmatic treatment guidance; (c) diagnostic stewardship (appropriate testing and interpretation); (d) documentation of indication, agent, and planned duration; (e) audit and feedback with benchmarking; (f) education and low-threshold access to materials; and (g) cross-sectoral communication across outpatient, inpatient. and emergency settings. Table 1 maps AnTiB activities to these principles.

## 5. Discussion and Conclusions

In 2016, the local, outpatient project AnTiB was founded by the physicians’ network “Ärztenetz Bielefeld” in order to establish improved communication about AMS and to help harmonize antibiotic prescribing. This resulted in a variety of activities, among them the development and implementation in Bielefeld of best-practice recommendations for pediatrics, gynecology, general medicine, urology, and otorhinolaryngology. These promote the idea that psychological and behavioral factors need to be taken into consideration if one wishes to advance the cause of AMS. By these means, AnTiB was able to gain broad acceptance among practitioners and emergency departments in Germany and was later able to transfer the locally developed guidelines to a broader audience: other regions of Germany.

AMS recommendations in Germany combine core stewardship principles (locally adapted treatment guidance, diagnostic stewardship, documentation of indication and duration, prospective audit and feedback, restriction of selected reserve agents, education and local monitoring) with operational measures tailored to care setting. The AnTiB consensus recommendations align closely with these core principles—for example, by promoting narrow-spectrum therapy, defined durations, watchful waiting, improved documentation, and diagnostic guidance—and are therefore consistent with hospital AMS objectives. Hospitals additionally implement setting-specific tools (e.g., routine in-house AMS team reviews) that are rarely applicable in primary care. By jointly developing guidance with inpatient colleagues, AnTiB helped harmonize prescribing across the outpatient–inpatient interface, while acknowledging that some hospital measures cannot be directly transferred to ambulatory practice.

Following the AnTiB example, the ABS-Network Bielefeld was established to further promote interdisciplinary exchanges, including outpatient physicians of different medical specialties, hospitals, laboratories, and pharmacies, and by implementing AMS into medical education at multiple levels. Meanwhile, AnTiB projects have also been integrated in the regional ABS-Network Westfalen-Lippe. The bottom-up approach embodied here emphasizes AMS integration, as well as scientific evaluation and publication.

AnTiB has additionally achieved the following: The group’s ability to address the problem of uncertainty and variability in antibiotic prescribing resulted in very positive responses from many medical specialties and settings, several of which reported reduced trust among patients before. This encouraged the establishment of local consensus recommendations for pediatrics, gynecology, general medicine, urology, and ENT in Bielefeld. These became supported not only by outpatient care providers, but also by emergency outpatient departments at local hospitals.

From these observations, we conclude that for the success of AMS interventions, both the establishment of social structures for communication and the quality of the communication itself are crucial. Framing inappropriate prescribing not as a personal shortcoming of individual prescribers but rather as a structural and shared issue of uncertainty among all professionals was shown to improve participation and acceptance. In 2019, a nationwide working group (ABSaP) was founded by German pediatricians to regularly update and disseminate national guidelines and training materials, as well as to support AMS activities nationwide by offering expert advice and resources.

One major limitation of our work is the absence of facility-level antibiogram data. Outpatient care is delivered by many independent physicians, and there is no centralized local resistance surveillance for the outpatient sector. Where resistance data were required to inform recommendations, we relied on regional and national surveillance data (for example, Robert Koch-Institute’ ARS (“Antibiotika-Resistenz-Surveillance”) [41]) and on clinical experience from collaborating laboratories. This limitation highlights a key gap for outpatient AMS: routine, local outpatient resistance surveillance is lacking but would substantially improve the tailoring of stewardship measures. We therefore recommend investment in regionally aggregated outpatient antibiograms or improved laboratory–primary care reporting as a priority for future AMS work.

Another limitation is the continuous lack of causal attribution of decreases in antibiotic prescribing to AnTiB. This is challenging, as prescribing is influenced by various factors (e.g., variations in infection dynamics, vaccination programs, care-seeking during the pandemic, other AMS activities). However, on a local level, AnTiB was associated with improved antibiotic prescribing [42]. Controlled evaluations (for example, interrupted time series with comparison regions or cluster-randomized designs) are required to quantify causal effects.

The developments described here demonstrate that it is possible to establish a cross-regional AMS network with a manageable amount of effort and that this may provide an opportunity to activate existing AMS potential in various specialist groups and institutions. The response to practical AMS activities was overall positive among healthcare professionals. Experiencing personal uncertainty and perceiving external factors as exerting a negative influence on prescribing decisions may be factors in these respects. Incongruity between professional and personal standards and actual prescribing behavior may result in a degree of cognitive dissonance. Participants frequently report that peer-to-peer discussions addressing the prevalence of such uncertainties are positive, stress-alleviating experiences.

To place the AnTiB experience in an international context, it is important to note that many of the behavioral drivers and intervention strategies described here are consistent with evidence from other countries. Inappropriate outpatient antibiotic use remains widespread [43], and randomized trials and implementation studies have shown that behavior-focused measures—communication skill training, clinician ‘nudges’ and social norm feedback, locally adapted guidance, and audit and feedback—can reduce unnecessary prescribing [20,44]. Major stewardship frameworks for outpatient care (for example, the CDC Core Elements [45] and the TARGET toolkit [46]) likewise prioritize clinician engagement, locally relevant guidance, and point-of-care support and feedback, mirroring core features of the AnTiB approach. These concordances suggest that the psychological foundation of prescribing behavior is broadly trans-cultural and that the core behavioral strategies used by AnTiB are likely to be relevant beyond Germany. At the same time, the precise design, implementation, and likely impact of outpatient AMS programs depend heavily on health system factors (regulation, reimbursement, diagnostic access, and primary care organization). Thus, while AnTiB can illustrate which elements merit inclusion and can stimulate international discussion, straightforward, unmodified transfer to other settings is not appropriate—local adaptation and evaluation are required.

Outpatient AMS initiatives should consider participant- and application-orientated perspectives in order to achieve high acceptance. Therefore, AMS recommendations should be available free of cost and with a low access threshold. Our review shows that a high level of acceptance and satisfaction may be achieved [7]. By deciding on a bottom-up approach with cross-sectional, intra- and interdisciplinary participation, AnTiB integrated local cultural aspects into its AMS implementation. By these means, its activities and outputs gained broad acceptance among practitioners and emergency departments, with its guidelines being adopted in multiple regions of Germany. Besides the practice guidelines, activities have included benchmark-type antibiotic prescription reports for individual practitioners, as well as analysis of the prescription dynamics in the region [42,47,48]. Published data underlines the effect of more conservative, guideline-concordant prescribing in the local Bielefeld setting after AnTiB implementation [42]. Local networking activities, training programs, and meetings were conducted, and information websites were established. The inclusion of cultural and behavioral aspects in AMS has been shown to be key to its effectiveness. The fact that the local project concept has continued to extend to other regions demonstrates the positive expansion potential of networks.

However, to achieve greater advancements in AMS, ongoing financial, structural, and non-material support is critical, as are standardized, nationwide feedback forms for outpatient care providers and antibiotic-specific pharmacotherapy advice for settings with high prescribing volumes. To further support AMS-structured training programs, innovative remuneration models for financial reward and secured public funding for AMS should be established.

## Figures and Tables

**Figure 1 antibiotics-14-01068-f001:**
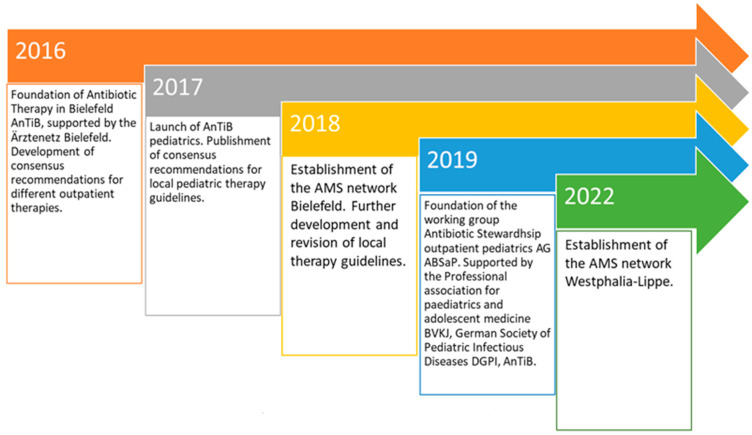
Timeline for (further) development and dissemination of the AnTiB project.

**Table 1 antibiotics-14-01068-t001:** AMS principles and measurements addressed by AnTiB and the ABS-Network Westphalia-Lippe. Information materials are available at https://www.uni-bielefeld.de/fakultaeten/gesundheitswissenschaften/ag/ag2/antib/index.xml (accessed on 5 September 2025) [37].

AMS Principles Addressed by AnTiB	AMS Measurements Addressed by AnTiB and the ABS-Network Westphalia-Lippe
Organizational principles at different levels: local, regional, federal	Locally consented recommendations developed into national guidelines
Addressing a variety of medical specialties	Locally consented guidelines in pediatrics, gynecology, general medicine, urology and ENT; national pediatric AMS guidelines anchored in local initiatives
Connecting different types of care settings: outpatient, inpatient, emergency care	Local and regional network structures
Early participation of and feedback from prescribers, respecting daily life working conditions	Training on AMS principles provided on a regular basis, through either in-person or online learning
Bottom-up approach	Peer-to-peer communication
Connecting practice with evidence-based science	Common networks and working groups of practitioners and scientific organizations
Social commitment	Formal consensus process and formal vote on consensus recommendations
Pragmatic reduction in complexity	Integrating AMS into the daily workflow with the use of a convenient, printed, coat-pocket-sized cards
Low-threshold access	Free of charge, various training programs
Audit and feedback	Feedback regarding individual antibiotic prescriptions for physicians
Sustainable network structures	Regional public relations work, e.g., for AMR Awareness Week
Information materials available free of charge	Integration of AMS principles into university education for public health and medical students at Bielefeld University
Evaluation	Regular local and regional evaluations of the frequency and selection of antibiotic prescriptions and participant surveys regarding satisfaction with AMS recommendations and their applicability

## Data Availability

Data sharing is not applicable.

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
