# Peer review of "Changing Antibiotic Prescribing Cultures: A Comprehensive Review of Social Factors in Outpatient Antimicrobial Stewardship and Lessons Learned from the Local Initiative AnTiB"

_antibiotics, 2025, doi:10.3390/antibiotics14111068_

Round 1

Reviewer 1 Report

Comments and Suggestions for Authors

QUESTION/COMMENTS

  1. What are the examples of reserved medications?
  2. "AB prescribing rates in the eastern and southern federal states are lower."  Why?  Do you mean prescribing "habits" are in line with the antibiotic stewardship?  Please clarify. 
  3. What are "specific" AMS recommendation in Germany and how much of it has consistency with the antibiotic stewardship measure in your hospital clinical wards? 
  4. Please further clarify bottom-up approach by providing specific clinical example. 
  5. "Incongruity between professional and personal standards."  This precisely this novel psychological discomfort applies to the overall concept of AMR and the stewardship programs.  Can application of the AI in part eliminate such a discomfort, although an example of discomfort can be helpful.  
  6. "A substantial proportion of health practitioners have reported significant dissatisfaction with the now improved antibiotic prescribing practice."  What percent of healthcare practitioners?  Where is the detail of now improved antibiotic prescribing practice in this paper?  
  7. The authors talk about antibiotic resistance, but do not provide example of misuse and overuse of antibiotics.  Please name three antibiotics with the highest rates of resistance in your healthcare facility.  
  8. What is the main take-home message of this paper and how it contributes to our current knowledge of AMR?  

Author Response

Dear Reviewer,

thank you for the questions and suggestions. We have provided our point by point response below:

1. What are the examples of reserved medications?

Line 52-55, original version: „… broad variations in global and local antibiotic prescribing practices have been observed for longtime, both in connection with the number of prescriptions given, (a quantitative measurement), as well as in relation to AB misuse, especially of "reserve" medications, (a qualitative measurement).“

Authors: Thank you for the reviewer’s comment — the term “reserve” indeed needs clarification. In Germany "the Federal Joint Committee" shall, upon request, exempt the pharmaceutical company from the obligation to submit the evidence required under paragraph 1, sentence 3, numbers 2 and 3 if the product is an antibiotic that is effective against infections caused by multidrug‑resistant bacterial pathogens for which only limited alternative therapeutic options are available, and if the use of this antibiotic is subject to strict indication (reserve antibiotic)." (source: Epidemiolog. Bulletin 35/2024 „Einstufung von Reserveantibiotika“ bzw. „Gesetzliche Verpflichtungen des RKI und BfArM im Rahmen der Einstufung von Reserveantibiotika“ https://www.rki.de/DE/Aktuelles/Publikationen/Epidemiologisches-Bulletin/2024/35_24.pdf?__blob=publicationFile&v=2)

The reviewer’s point is correct: this restriction is overly specific. “Reserve” refers to a relatively narrow group of antibiotics, which was not intended here. For our purposes — and particularly for the outpatient sector that this article addresses — the term “broader‑spectrum antibiotics” is more appropriate. We therefore changed the passage to: "... especially broader‑spectrum antibiotics." (line 57, revised version)

Examples: Amoxicillin-Clavulanic acid, Cephalosporins, Macrolides, Fluorchinolones

2. "AB prescribing rates in the eastern and southern federal states are lower."  Why? 

Line 59-61, original version: „Within Germany, AB prescribing rates in the eastern and southern federal states are lower, while higher rates persist in the western areas of the country [5,6].“

Authors: To date, this phenomenon has not been satisfactorily explained. Even the most detailed investigation (Scholle et al. 2020 [6]) also offers no explanation. One possible reason may be variation in local or regional prescribing habits. Further research would be promising, since identifying these local and regional factors could allow them to be adopted as best practices in other areas. We did include the information of the lacking explanation for this phenomenon in our revision (line 63-64, revised version).

Do you mean prescribing "habits" are in line with the antibiotic stewardship?  Please clarify. 

Authors: Thank you for this request for clarification. We do believe this question belongs to the sentence „Both the AnTiB and ABS Network Westphalia-Lippe work principally aim to address the changing of habits, the resolution of uncertainties.“ (line 260-61, original version) We agree that local prescribing “habits” aren’t already in line with antibiotic stewardship (AMS/ABS) and that this sentence is misleading. On the contrary, habits often develop locally and can diverge from AMS principles. The AnTiB project and the ABS Network Westphalia‑Lippe both explicitly aim to identify these local/regional prescribing patterns and to align them with AMS/ABS through locally adapted guidance, education, decision support and quality‑improvement measures. We have clarified the text as shown below (Line 269-74, revised version): "Both the AnTiB project and the ABS Network Westphalia‑Lippe primarily aim to change local prescribing habits and reduce diagnostic and therapeutic uncertainty by aligning routine practice with antibiotic stewardship principles. To achieve this, the projects promote locally adapted consensus guidance, easy‑to‑use decision support and accessible best‑practice information, together with education and quality‑improvement measures."

3. What are "specific" AMS recommendation in Germany and how much of it has consistency with the antibiotic stewardship measure in your hospital clinical wards? 

Authors: Thank you for this request for clarification. In Germany, specific AMS recommendations comprise a set of evidence‑based measures (see list below) that are endorsed in national frameworks and specialty guidelines. The AnTiB recommendations are largely consistent with these core principles and overlap substantially with hospital AMS activities where the outpatient–inpatient interface is concerned, while hospital wards additionally apply several inpatient‑specific interventions that are not feasible in ambulatory care. We have added a short paragraph to the discussion summarizing these points and giving concrete examples of alignment and differences (line 334-345, revised version).

Quick summary: what “specific” AMS recommendations in Germany include

  • Establish multidisciplinary AMS teams with defined responsibilities (clinical, microbiology, pharmacy, nursing, infection prevention).
  • Develop and maintain locally adapted therapy guidelines (empirical choices, dose, duration) and formularies.
  • Restriction or controlled use of certain “reserve”/critical antibiotics and pre‑authorization for selected agents.
  • Requirements for documentation of antibiotic indication, dose and planned duration on the medication order/prescription.
  • IV to oral switch and de‑escalation strategies where clinically appropriate.
  • Regular monitoring of antibiotic consumption and resistance surveillance with feedback to clinicians.
  • Continuing education/training and communication skills for prescribers.

Note: these items are reflected in national guidance documents and in KRINKO/RKI and specialty guidance; we can add specific citations on request.

How much consistency with hospital ward AMS measures?

Authors: We do achieve a high concordance on core principles: AnTiB’s recommendations (narrowest effective spectrum, shortest effective duration, “watchful waiting” where appropriate, clear documentation of indication/dose/duration, and emphasis on diagnostic and communication strategies) directly mirror the central goals of hospital AMS. These shared elements support harmonization across the outpatient–inpatient interface. Because AnTiB developed recommendations jointly with hospital clinicians, the resulting guidance improved consistency at transitions of care (e.g., emergency departments, hospital discharge prescriptions), reducing conflicting messages between different settings.

4. Please further clarify bottom-up approach by providing specific clinical example. 

Authors: Thank you for pointing out the repeated use of “bottom‑up.” We agree this needs clarification. By "bottom‑up", we mean an approach that is initiated and driven by outpatient practitioners themselves, rather than imposed from specialty societies or central bodies. In the AnTiB project, local groups of practicing physicians (for example, paediatricians) convened to develop antibiotic‑stewardship (AMS/ABS) recommendations tailored to the realities of primary care, motivated by the goal to achieve harmonised communication and therapeutic strategies following best practice. All recommendations were developed by local outpatient physicians, aligned with the evidence through literature review and expert consultation, and continuously informed by frontline practitioners. A detailed description of this consensus process exemplary for the paediatric process is provided in Soler Wenglein et al. (Eur J Pediatr. 2025;184(2):149. doi:10.1007/s00431-024-05964-y). We have added a sentence to clarify what is meant by "bottom‑up" in line 110-14, revised version.

5. "Incongruity between professional and personal standards."  (line 347, original version) This precisely this novel psychological discomfort applies to the overall concept of AMR and the stewardship programs.  Can application of the AI in part eliminate such a discomfort, although an example of discomfort can be helpful.

Authors: We thank the reviewer for highlighting the psychological discomfort that can arise when clinicians’ personal or situational pressures conflict with professional standards. Although AnTiB does not yet include AI interventions and our team lacks specific AI expertise, we agree that AI has promising roles in reducing such discomfort (for example, by providing patient‑specific decision support, communication templates for difficult conversations, automated documentation of rationales, and AI‑driven audit and feedback). At the same time, AI is associated with important concerns (accuracy, bias, transparency, overreliance, and regulatory issues) that require rigorous evaluation. As we do not have any AI experience in medical practice yet, we did not include these considerations into our work. But we do agree this will be an important aspect for future works.

6. "A substantial proportion of health practitioners have reported significant dissatisfaction with the now improved antibiotic prescribing practice."  What percent of healthcare practitioners? 
Line 344-45, original version: „A substantial proportion of practitioners have reported significant dissatisfaction with the now improved antibiotic prescribing practices. “

Authors: Thank you for highlighting this unclear statement. We agree that the sentence is misleading because it is only supported by unpublished data. We have removed the sentence from the manuscript (line 381-82, revised version).

Where is the detail of now improved antibiotic prescribing practice in this paper?

Authors: Thank you for this question. The details of antibiotic prescribing practice and an analysis of changes after implementation of AnTiB have been published in German (including an English abstract) elsewhere: Bornemann et al. Analyse von Einflussfaktoren auf ambulante pädiatrische Antibiotikaverordnungen in Bielefeld 2015–2018 [Analysis of factors influencing outpatient paediatric antibiotic prescriptions]. 2024). We have added a reference for these results in our discussion (line 415-17, revised version).

7. The authors talk about antibiotic resistance, but do not provide example of misuse and overuse of antibiotics.  Please name three antibiotics with the highest rates of resistance in your healthcare facility.  

Authors: Thank you for this request. We cannot provide facility‑level antibiogram data for the outpatient sector because outpatient care in our region is fragmented across many independent practices and there is no routine, central resistance monitoring at the local outpatient level. For this reason, we did not report any single‑facility “top three” resistance rates. Instead, our stewardship recommendations were informed by regional and national surveillance and by local clinical experience. To address the reviewer’s concern, we have added text citing surveillance tools (e.g., RKI ARS „AntibiotikaResistenz-Surveillance“) and giving representative examples of commonly observed resistance problems in community-acquired pathogens that are relevant to outpatient prescribing. Information on antibiotic resistance in Germany is available for the outpatient sector via the Robert Koch-Institute‘s ARS project; https://amr.rki.de/Content/Datenbank/ARS/ResistanceOverview.aspx) and can be checked for sector, region (we are north/north-west) for the last years since 2008. Examples for common outpatient relevant bacteria are:

  • Outpatient Streptococcus pneumoniae in west/north-west Germany in 2024: resistant for doxycycline in 14,8% and for erythromycin 12,7%.
  • Outpatient Escherichia coli in west/north-west Germany in 2024: resistant for amoxicillin/clavulanic acid in 27,9% and for cefuroxime in 13,4%
  • Outpatient Staphylococcus epidermidis in west/north-west Germany in 2024: resistant for doxycycline in 21,8 % and for amoxicillin/clavulanic acid in 39,8%

We have included this limitation in line 367, revised version, and following.

8. What is the main take-home message of this paper and how it contributes to our current knowledge of AMR?  

Authors: Thank you for this helpful question. The main take‑home message of our report is that sustainable improvements in outpatient antibiotic prescribing require explicit incorporation of behavioural and social‑science insights into AMS design and implementation. By documenting contextual drivers of prescribing and describing the clinician‑led, bottom‑up AnTiB initiative, we demonstrate a practical model for how locally adapted consensus processes, peer networks, and behaviour‑focused interventions (communication training, social‑norm feedback, easy decision aids and audit/feedback) can increase acceptance and alignment with stewardship principles. This paper thus contributes to AMR knowledge by (1) emphasising the trans‑cultural psychological determinants of prescribing, (2) providing a replicable, practitioner‑driven implementation approach, and (3) identifying key gaps for future research and system investment (e.g., local surveillance, robust outcome evaluation, and scalable support mechanisms).

With best regards and on behalf of the co-authors

Roland Tillmann

Reviewer 2 Report

Comments and Suggestions for Authors

I would like to thank the editors for entrusting me with the revision of this review article.

The problem of increasing antibiotic resistance is widely known. Expanding knowledge about programs and initiatives that bring about positive change in this area is very valuable. For this reason, I consider the presented work to be valuable material for disseminating information about such programs, such as the AnTiB program mentioned in the work.

I have only two suggestions aimed at improving the readability of this work:

I suggest expanding the section of the abstract concerning the conclusions drawn from the presented work

Line 226 contains the abbreviation MRGN, which is only explained in the Abbreviations list. I suggest explaining the abbreviation also in the place where it is used in the text

Author Response

Dear reviewer,

thank you for your helpful comments and suggestions. Please find our point-by-point response below.

1. I suggest expanding the section of the abstract concerning the conclusions drawn from the presented work

Authors: Thank you for this helpful suggestion. We have expanded the abstract’s conclusion as requested. The final sentence, we have added (placed at the end of the Abstract, line 40-43, revised version) reads:

"Integrating behavioural and social‑science approaches into outpatient antimicrobial stewardship — exemplified by the practitioner‑led AnTiB model — improves acceptability and alignment with stewardship principles; wider adoption will require local adaptation, routine outpatient resistance surveillance, structured evaluation, and sustainable support."

2. Line 226 contains the abbreviation MRGN, which is only explained in the Abbreviations list. I suggest explaining the abbreviation also in the place where it is used in the text

Authors: We have changed this part to „(e.g., oral cefuroxime due to poor oral bioavailability and potential development of multi-drug resistant gram-negative bacteria (MRGN)).“

With best regards and on behalf of the co-authors

Roland Tillmann

Reviewer 3 Report

Comments and Suggestions for Authors

This paper reviews the psychological and behavioral factors that influence antimicrobial stewardship (AMS) implementation, with a focus on outpatient care where most antibiotics are prescribed. Its main contribution is the presentation of the “Antibiotic Therapy in Bielefeld” (AnTiB) initiative, a bottom-up, cross-sectoral approach that fosters rational prescribing through locally developed guidelines, interdisciplinary collaboration, and regular training. The strengths of the manuscript lie in its comprehensive analysis of prescribing behavior, its demonstration of how organizational and psychosocial aspects can be integrated into AMS, and its evidence of successful local implementation that has expanded into regional and national frameworks. It is particularly important that emphasis is placed on locally developed recommendations, since without local data on antibacterial resistance, successful infection therapy cannot be achieved.

The study’s relevance seems primarily local - national, as it focuses on an intervention conducted only in Germany. In addition, the discussion does not sufficiently compare the results with similar studies from other countries, which restricts its broader contribution to understanding the psychological and behavioral factors influencing antimicrobial stewardship (AMS) and the potential applicability of bottom-up approaches in diverse contexts. To enhance the significance and generalizability of the findings, the discussion should include a more thorough comparison with international research.

Author Response

Dear reviewer,

thank you for your helpfull suggestion and your time. Please find our answer to your main suggestion below:

The study’s relevance seems primarily local - national, as it focuses on an intervention conducted only in Germany. In addition, the discussion does not sufficiently compare the results with similar studies from other countries, which restricts its broader contribution to understanding the psychological and behavioral factors influencing antimicrobial stewardship (AMS) and the potential applicability of bottom-up approaches in diverse contexts. To enhance the significance and generalizability of the findings, the discussion should include a more thorough comparison with international research.

Authors: Thank you for this important suggestion. The manuscript was prepared in response to a request specifically asking for a report on the AnTiB initiative, which is why the paper focuses on the local and regional experience in Germany. We agree that situating our findings within the international literature strengthens generalizability. We have expanded the discussion to compare our results with international AMS work and added citations to major studies (line 391-95, revised version, referencing Fleming‑Dutra KE et al., JAMA 2016; Meeker D et al., JAMA Intern Med 2016; Arnold SR & Straus SE, Cochrane Database Syst Rev 2005; CDC Core Elements and the TARGET toolkit).

We would like to emphasize two related points we have now added to the discussion: (1) the psychological foundations of behaviour are broadly comparable across cultures, so the barriers and behavioural strategies we describe have international relevance; and (2) the precise design and feasibility of an outpatient AMS programme depend heavily on the organisation of the local health system (regulatory context, diagnostics, reimbursement, primary‑care structures). Thus, AnTiB can illustrate which elements should be considered for inclusion in AMS programmes and can stimulate international discussion, but a direct, unmodified transfer to other countries is not appropriate — findings should be discussed and adapted to local conditions. 

With best regards and on behalf of the co-authors

Roland Tillmann

Round 2

Reviewer 1 Report

Comments and Suggestions for Authors
  1. AnTiB is not clearly explained.  What is the aim?  What was the basis of this project? Please elaborate more about this project and how this project contributes to our overall understanding of the antibiotic challenges?
  2. "The consensus recommendations produced by the project aim to reduce antibiotic prescriptions and are based on simple AMS principles."  Do you mean AnTiB project?  Where is the delineation of AMS principles in the manuscript.  
  3. What is the view of the German Society for Pediatric Infectious Diseases about this study?  Have they been consulted for this project? 
  4. Recent reports from Germany indicate a decline in overall antibiotic prescribing habits in children.  Is this decline due to AnTiB project?
  5. Please list three pathogens in pediatric ID that are associated in significant antibiotic resistance in endemic areas of Germany.     

Author Response

Reviewer:

  1. AnTiB is not clearly explained.  What is the aim?  What was the basis of this project? Please elaborate more about this project and how this project contributes to our overall understanding of the antibiotic challenges?
  2. "The consensus recommendations produced by the project aim to reduce antibiotic prescriptions and are based on simple AMS principles."  Do you mean AnTiB project?  Where is the delineation of AMS principles in the manuscript.  
  3. What is the view of the German Society for Pediatric Infectious Diseases about this study?  Have they been consulted for this project? 
  4. Recent reports from Germany indicate a decline in overall antibiotic prescribing habits in children.  Is this decline due to AnTiB project?
  5. Please list three pathogens in pediatric ID that are associated in significant antibiotic resistance in endemic areas of Germany.

Authors:

We thank the reviewer for the constructive comments. Below we address each point and describe the manuscript changes.

1) Reviewer comment: "AnTiB is not clearly explained..."

Response: We have added a concise "AnTiB — project profile" paragraph to Section 3 that describes aims, target groups, participants, development process and intended contributions to outpatient AMS (new text, line 183). Evaluation data are unpublished for user-satisfaction and published for overall antibiotic use in Bielefeld, as referenced in line 408 and 450.

2) Reviewer comment: "Where are the delineated AMS principles?"

Response: We have added an explicit list of core AMS principles operationalized by AnTiB (new paragraph in Section 3) and cross‑referenced Table 1 (line 110 and 331).

3) Reviewer comment: "What is the view of the DGPI?"

Response: We have clarified the relationship: the AnTiB pediatric activities are in line with the goals of the DGPI which supports national dissemination and training activities. (See Section 4, new paragraph.). The outpatient pediatric recommendations of AnTiB and DGPI are aligned and have been jointly published in English as referenced in line 295 and following.

4) Reviewer comment: "Is the national decline due to AnTiB?"

Response: We cannot attribute nationwide declines to AnTiB; we have clarified this in the limitations section and propose controlled evaluations to assess impact. We added a reference for the local decline, that seems to be associated with AnTiB (but no causal attribution is possible due to the various factors influencing antibiotic prescribing) (line 399 and following).

5) Reviewer comment: "List three pediatric pathogens..."

Response: Thank you for this suggestion. While pathogens such as Escherichia coli (notably increasing rates of ESBL‑producers), Klebsiella pneumoniae (ESBLs and occasional carbapenemases) and Streptococcus pneumoniae (serotype‑dependent reduced penicillin susceptibility) are clinically relevant in outpatient pediatrics in Germany, a detailed overview of pathogen‑specific resistance epidemiology falls outside the implementation‑focused scope of this manuscript. For current, comprehensive resistance data we therefore refer readers to national surveillance sources (Robert Koch-Institute ARS) and did not include this information in our article.

Best regards

Reviewer 3 Report

Comments and Suggestions for Authors

Dear Authors,

Thank you for your thoughtful and comprehensive revision. The expanded discussion and integration of international studies have substantially strengthened the manuscript’s generalizability and contextual relevance. I also appreciate the clarification regarding cross-cultural applicability and system-specific adaptation. I am satisfied with the revised version of the manuscript.

Author Response

Reviewer: Dear Authors,

Thank you for your thoughtful and comprehensive revision. The expanded discussion and integration of international studies have substantially strengthened the manuscript’s generalizability and contextual relevance. I also appreciate the clarification regarding cross-cultural applicability and system-specific adaptation. I am satisfied with the revised version of the manuscript.

Authors: Dear Reviewer, thank you very much for your positive feedback and your support in improving the manuscript. Best regards!

Round 3

Reviewer 1 Report

Comments and Suggestions for Authors

The manuscript the manuscript has been adequately improved for publication in Antibiotics